# Analysis of Dormancy-Associated Transcriptional Networks Reveals a Shared Quiescence Signature in Lung and Colorectal Cancer

**DOI:** 10.3390/ijms23179869

**Published:** 2022-08-30

**Authors:** Adriano Cuccu, Federica Francescangeli, Maria Laura De Angelis, Alessandro Bruselles, Alessandro Giuliani, Ann Zeuner

**Affiliations:** 1Department of Oncology and Molecular Medicine, Istituto Superiore di Sanità, Viale Regina Elena 299, 00161 Rome, Italy; 2Environment and Health Department, Istituto Superiore di Sanità, Viale Regina Elena 299, 00161 Rome, Italy

**Keywords:** quiescence, dormancy, lung cancer, colorectal cancer

## Abstract

Quiescent cancer cells (QCCs) are a common feature of solid tumors, representing a major obstacle to the long-term success of cancer therapies. We isolated QCCs ex vivo from non-small cell lung cancer (NSCLC) and colorectal cancer (CRC) xenografts with a label-retaining strategy and compared QCCs gene expression profiles to identify a shared “quiescence signature”. Principal Component Analysis (PCA) revealed a specific component neatly discriminating quiescent and replicative phenotypes in NSCLC and CRC. The discriminating component showed significant overlapping, with 688 genes in common including ZEB2, a master regulator of stem cell plasticity and epithelial-to-mesenchymal transition (EMT). Gene set enrichment analysis showed that QCCs of both NSCLC and CRC had an increased expression of factors related to stemness/self renewal, EMT, TGF-β, morphogenesis, cell adhesion and chemotaxis, whereas proliferating cells overexpressed Myc targets and factors involved in RNA metabolism. Eventually, we analyzed in depth by means of a complex network approach, both the ‘morphogenesis module’ and the subset of differentially expressed genes shared by NCSLC and CRC. This allowed us to recognize different gene regulation network wiring for quiescent and proliferating cells and to underpin few genes central for network integration that may represent new therapeutic vulnerabilities. Altogether, our results highlight common regulatory pathways in QCCs of lung and colorectal tumors that may be the target of future therapeutic interventions.

## 1. Introduction

In mammalians, most tissues and organs contain small populations of quiescent stem cells that serve as a reservoir during cellular turnover and upon tissue damage [1]. Likewise, many tumors contain subpopulations of quiescent/slow cycling cells involved in clinically relevant steps of tumor development such as chemo-resistance, metastatic dissemination, dormancy and relapse [2,3]. In the last decade, there has been a progressive understanding of QCCs importance for cancer therapy. Consequently, the number of studies focused on QCCs and tumor dormancy increased exponentially, highlighting the complexity of the quiescent state. In fact, quiescence is a heterogeneous state, both in normal tissues and in cancer [4,5]. First, different levels of quiescence depth can be found in normal and possibly in neoplastic cells [6,7,8,9]. Second, quiescent cancer cells are found in different moments of the tumor’s lifetime such as before, during and after treatments; during latency; or in actively growing cancers. Third, QCCs can reside in different intra-tumor or extra-tumor locations, thus being subject to a different spectrum of microenvironmental signals [3]. The peculiar behavior of ‘regime change’ from quiescent to proliferative state of QCCs stimulated biophysical studies along the line of non-linear dynamics approach [10,11] that in turn opened new general perspectives on cancer development [12]. New insights into the mechanisms driving cancer cell quiescence have led to the development of new experimental models of cancer cell dormancy in vitro and in vivo [4] and in the identification of new dormancy-specific therapies that are currently under clinical evaluation [13]. In fact, therapies targeting QCCs would be useful in several clinical circumstances, i.e., to overcome chemo/radioresistance and to prevent metastatic dissemination, tumor reawakening and relapse. Dormancy-specific therapeutic strategies include drugs that target a plethora of cellular processes including autophagy, stress signaling, nuclear receptors, adhesion signaling, metabolism and epigenetic regulation [14]. Clearly, the possibility to identify effective therapies directed against QCCs depends on the identification of QCCs-specific vulnerabilities. The identification and isolation of QCCs has been previously performed through different methods, such as the histone H2B-GFP approach or through labelling techniques, including PKH26, bromodeoxyuridine or carboxyfluorescein succinimidyl ester. PKH26 is a dye that incorporates into the membrane of cells and is equally distributed to daughter cells when they divide [15]. PKH26 labelling was used to isolate QCCs from breast cancer [16], colorectal cancer [17,18], glioblastoma [19,20] and osteosarcoma [21]. QCCs populations obtained through PKH26 labelling contain tumor cells negative for the proliferation marker Ki67 [22] and are enriched in cells with enhanced cancer initiation potential, expression of “stemness” genes and resistance to chemotherapy [16,18,20,21,22]. Starting from the hypothesis that QCCs relative to different tumors may share a core of epigenetic regulators responsible for the specification of the quiescent/slow cycling state, we compared QCCs gene expression profiles of quiescent/slow cycling cells with the profiles of rapidly proliferating cells separately in NSCLC and CRC. Then, we performed a cross-analysis of genes and pathways differentially expressed in QCCs of NSCLC and CRC, identifying a signature of quiescence-associated genes and pathways common to both tumors. On a more theoretical perspective, we analyzed in detail the morphogenesis module in quiescent and proliferating cells of both tumors, pointing to relevant differences in gene regulatory network structure. The network approach both indicated ‘high centrality’ genes that differentiate quiescent and proliferative status and a wiring architecture of gene expression correlation for quiescent cells state more fit to a metastable state like dormancy with respect to actively proliferating cells [23]. These results show for the first time the existence of common traits that characterize QCCs of lung and colorectal cancer, providing new insights into the mechanisms responsible for quiescence regulation in solid tumors.

## 2. Results

### 2.1. Data Generation and Analysis

NSCLC and CRC specimens were obtained from patients undergoing surgical resection and used to generate multicellular spheroid cultures that were validated by genomic and proteomic analyses, marker expression and ability to reproduce phenocopies of the parental tumors when inoculated in immunocompromised mice [3,22,24,25,26]. NSCLC and CRC spheroids were stained with the lipophilic membrane dye PKH26, which is rapidly lost by actively proliferating cells and retained by slowly proliferating/quiescent cells. Bulk populations of PKH26-stained spheroid cells were inoculated subcutaneously into immunocompromised mice, and tumors were allowed to grow for three or four weeks (respectively for CRC and NSCLC), until the percentage of PKH26-positive cells reached approximately 5% of total tumor cells. At that time, tumors were collected and PKH26^+^ were separated from PKH26^−^ cells by fluorescence-activated cell sorting (FACS) upon additional labeling with anti-human EpCAM antibody, in order to avoid contamination with non-tumor cells. PKH26-positive and -negative fractions were lysed immediately after sorting and used for gene expression analysis as described in the Methods Section. Having obtained profiles of differentially expressed genes in colon and lung QCCs as compared to their actively proliferating counterpart, we asked whether we could identify a sub-set of genes, common to the two tumors, discriminating quiescent and replicative cell states. To do this, we adopted a largely unsupervised, data-driven strategy. This was an almost essential choice to escape the ‘curse of dimensionality’ with the consequent risk of chance correlation stemming from the very high number of gene products observed in both colon and lung experiments. Thus, we adopted a strategy based on Principal Component Analysis (PCA) as applied to the data set having different samples (gene expression profiles) as variables and gene products as statistical units. Only *a posteriori* the loading space generated by PCA was checked for the presence of one component exactly matching the PKH26^+^/PKH26^−^ in terms of loading values [27], thus collapsing the number of inferential tests to three-four. After having found a component that discriminates with no error and in completely unsupervised way the two groups, we stepped into the analysis of component scores identifying the genes with higher absolute scores, thus being ‘most representative’ of the discriminating component. Out of this selection, we focused on ‘high score’ elements shared by the two colon and lung tissues. These shared genes correspond to the most relevant genes discriminating PKH26^+^/PKH26^−^ conditions across different tissue contexts and were analyzed in terms of putative pathways they pertain. In parallel to the unsupervised PCA approach, we carried out a differential expression analysis of PKH26^+^/PKH26^−^ datasets with the supervised *limma* (Linear Models for MicroArray Data) approach in order to get a consistent extraction of relevant genes. *Limma* provides a measure of the differential expression between the two quiescent and replicative states for each gene, the log2 fold change (logFC). *Limma* trend is indeed a standard and powerful approach to detect significantly differentially expressed genes, developed to work with both Microarray and RNA-seq data [28]. Furthermore, focusing on the first 6000 genes and the last 6000 of the ranking built by sorting them in descending order by logFC in the colon and lung experiments, we performed Gene Set Enrichment Analysis (GSEA) and reported the significantly enriched pathways shared by the two tissues. These common pathways correspond to the most relevant ones characterizing the two PKH26^+^ and PKH26^−^ conditions regardless of the type of tissue. The consistency between unsupervised (deriving from PCA) and supervised (*limma*) gene selection demonstrated the consistency of the results. Finally, we applied complex network analysis of differentially expressed genes to the morphogenesis-related gene subset, which was differentially expressed by lung and colorectal QCCs.

### 2.2. PCA and Limma Trend Analysis of Lung Cancer Quiescent and Proliferating Cells

Starting from the assumption that quiescence dynamics are different in cancer models in vitro and in vivo, and that in vivo tumor models likely reproduce more faithfully the features of QCCs naturally present in human cancer, we used molecularly annotated 3D cultures of primary NSCLC cells [24,29] to generate subcutaneous tumor xenografts in immunocompromised mice. Cells were stained with PKH26 before inoculation, and PKH26^+^/PKH26^−^ tumor cells were collected by flow cytometry after three weeks of tumor growth. Gene expression profiles were generated by RNA-sequencing and the results were analyzed in parallel by PCA and *limma* analysis. PCA on log-transformed original expression data gave rise to a three-component solution explaining around 67% of total variance. Each component conveys an immediate meaning in terms of loading pattern (Table 1A,B):

Having assessed the presence of a specific (PC3) component discriminating, in a totally unsupervised way, the two PKH26^−^ and PKH26^+^ phenotypes (Figure 1A), we went in depth into the most discriminating genes by means of a *limma* trend analysis [28]. The analysis was performed on log counts per million and gave rise to the results reported in Figure 1B: the genes sorted by the log fold changes in decreasing order are reported in Appendix A. The first and the last 6000 genes were then compared with the first and the last 6000 genes of the ranking built on the scores on PC3 of the PCA in terms of their score on PC3 (top 6000 genes = genes characterizing PKH26^+^, last 6000 genes = genes characterizing PKH26^−^).

The *limma* result shares the 75% of the most up-regulated genes and the 72% of the most down-regulated genes with PCA (Figure 1C,D). This is a remarkable finding because it highlights a quite high consistency between the standard *limma* and unsupervised PCA results, giving us greater confidence on the reliability of the *limma* output.

Finally, we asked whether the quiescent cell population was characterized by an increased expression of stemness-related factors. Indeed, *limma* analysis of differentially expressed genes indicated an increased expression of four genes related to cancer stemness: the pluripotency factor KLF4, the PLAUR gene encoding for the urokinase plasminogen activator surface receptor (uPAR), CD44 and CD166 (Table 2). These findings indicate that, in lung cancer, there is at least a partial overlapping between quiescent label-retaining cells and cancer stem cells (CSCs).

### 2.3. PCA and Limma Trend Analysis of Colorectal Cancer Quiescent and Proliferating Cells

CRC cells were isolated from tumor xenografts generated with PKH26-stained multicellular spheroid cultures and gene expression profiles were obtained as described in our previous studies [22]. PCA on expression data gave rise to a four-component solution explaining around 99.5% of total variance. At odds with lung case, due to the lack of relevant information about line of origin or other relevant phenotypic features, we can only assign a meaning to PC1 and PC4 (Table 3A,B). PC1 corresponds (consistently with lung) to the ‘tissue attractor effect’, while PC4 exactly discriminates the two quiescent and proliferative phenotypes.

In this case, thanks to the more homogeneous biological material, the ‘tissue attractor’ component (PC1) is much more relevant than in the lung case, accounting for 98.7% of variance. All the samples have loading of the same sign on this component (the fact the sign is negative only depends on computational details, given that the directionality of the eigenvector is arbitrary) so it can be considered as a size component. The discrimination between PKH26^+^ and PKH26^−^ phenotypes stems from the fourth component (PC4) in which the two groups have opposite sign loadings (Table 3B, Figure 2A).

The comparison between PCA and *limma* results for the colon case is reported in Figure 2.

Having assessed the presence of a specific (PC4) component discriminating the two PKH26^−^ and PKH26^+^ phenotypes (Figure 2A), we focused on the most discriminating genes by means of a *limma* trend analysis [28], which gave rise to the results reported in Figure 2B: the genes sorted by the log fold changes in decreasing order are reported in Appendix A. The *limma* result shares 69% of the most up-regulated genes and 58% of the most down-regulated genes with PCA (Figure 2C,D). This finding is consistent with the results found in NSCLC.

Finally, we asked whether also in the case of colorectal cancer quiescent cells were characterized by an increased expression of stemness-related factors. Among the top genes emerging from limma analysis, we found four genes particularly related to cancer stemness and CRC CSCs: besides the pluripotency factor KLF4, we found three genes were specifically related to colorectal cancer stemness and self renewal, i.e., AXIN2, LGR5 and BMI1 (Table 4). These findings indicate that in colorectal cancer label-retaining quiescent cancer cells are enriched in CSCs.

### 2.4. Comparison between Genes Involved in the Quiescent State in Lung and Colon Cancer

In order to get a full picture of the quiescent/proliferative discrimination, it is important to focus on the discriminating genes shared by the two tissues (Figure 3).

The differential expression analysis of genes shared by lung and colon QCCs showed a partial overlapping of the two transcriptional programs, with approximately 15.8% of genes upregulated in lung cancer QCCs and 21.3% of genes upregulated in CRC QCCs, for a total of 688 genes shared between QCCs of both tumors (Figure 3A). Genes downregulated in QCCs of both tumors (599 in total) include approximately 14% of quiescence-associated genes for lung cancer and 17.1% for CRC (Figure 3B). KEGG pathway selection was applied to the 688 genes representing the “core transcriptional program” of QCCs. The results are shown in Figure 3C and highlight and increased involvement of pathways implicated in angiogenesis, EGFR and RAS signaling, integrins, TGF-β and WNT signaling, suggesting a central role of these pathways in solid tumor quiescence. 

Pathway Analysis was performed through the GSEA software (version 4.2.2, UC San Diego and the Broad Institute, San Diego, CA, USA) according to the gene-pathway matching algorithm reported in [32]. The application of GSEA to the *limma*-ranked results gave rise to the significant pathways (nominal *p*-value < 0.05) shared by the two lung and colon tissues shown in Table 5. The shared significantly enriched pathways between the two tissues are shown in Appendix A for the quiescent state and in Appendix A for the replicative one. The set of all significantly enriched pathways in the two tissues is reported in Appendix A for the lung system and in Appendix A for the colon one.

Table 5 reports the pathways shared by the two lung and colon systems that differentiate quiescent and replicative phenotypes. These shared pathways can be thus considered as proxies of the ‘tissue independent’ possible mechanisms governing the quiescent/replicative transition.

Figure 4 reports the enrichment plots of some interesting common pathways significantly enriched in the quiescent condition between the lung and colon tissues, and the Venn diagrams of the enriched pathways in the two PKH26^+^ and PKH26^−^ states.

Overall, the lung system shares almost 30% of the significantly enriched pathways in either state with the colon system. This finding leads us to state that the common trait of the quiescent/replicative transition between the two tissues is not negligible.

### 2.5. Network Analysis in Quiescent and Proliferating Cancer Cells (with a Specific Emphasis on Morphogenesis Module)

In order to get some hints on the gene expression regulation network constituted by the ‘transcriptional core’ discriminating quiescent and proliferative states in both tissues, we performed a co-expression network analysis.

The existence of a strong correlation among differentially expressed genes shared by the two tissues is the image in light of a coordinated gene regulation network in charge of the transition between the two states. The gene expression network has genes as nodes: an edge between two nodes is drawn (and consequently the two genes are considered to interact) if their correlation coefficient exceeds the threshold of r = |0.7|.

Figure 5A shows the network built on the 688 genes up-regulated in the PKH26^+^ state shared by the lung and colon systems, considering the PKH26^+^ samples of both tumors. Appendix A reports relevant information on the interactions between the nodes of this network. Appendix A reports the betweenness centralities and the clustering coefficients of the nodes and these measures are displayed in Figure 5B. Appendix A shows the centrality scores for the genes in the TGF-β pathways. Appendix A and Figure 5B refer to the network built considering the usual threshold of Pearson r = |0.7|, while Appendix A and Figure 5A refer to a more stringent threshold r = |0.98|, which is useful to get a graphical picture of the network and, still more important, highlights the quasi-deterministic character of the gene regulation network underlying quiescent/proliferative transition.

In Figure 5A, is worth noting that ZEB2 occupies a central position in the same network domain as c-FOS, consistently with its role in stem cell regulation, plasticity and proliferation transition. Looking at the network, it is evident to note the extremely high density of edges, corresponding to a strictly coordinated behavior of the entire system. This density goes hand-in-hand with extremely elevated Clustering coefficient values (Clustering coefficient corresponds to the relative frequency of ‘complete triads’, which holds the condition: *if gene A is correlated with gene B and gene B is correlated with gene C, then even gene A is directly correlated with gene C*). While the Clustering coefficient has to do with the local network structure, betweenness deals with the long range signal transmission throughout the co-expression network. Betweenness of a node corresponds to the number of ‘shortest paths’ traversing it, in other words, a “high betweenness” gene corresponds to a gene involved in many signal transmission pathways across the network.

The usefulness of centrality indicators in gene co-expression networks have been reported in various application domains, such as cancer research [33], neuroscience [34] and protein science [35]. For instance, genes exhibiting high betweenness centrality scores have been proposed as candidate targets in different human and animal models [36].

The betweenness centrality encapsulates the property of node i as a bridging node in the network, corresponding to the number of shortest paths connecting any two nodes, j and k, which pass through node i. Nodes with large betweenness-centrality values are often called “high traffic” nodes [36].

Figure 5B projects the genes in the Clustering coefficient/betweenness space. In Figure 5B, the genes in the TGF-β pathways are coded in blue, and they occupy a specific location in the graph where they are easily recognizable (and not scattered around) consistently, pertaining to a specific pathway. The most striking peculiarity of these group of genes is the singularity of JAK1 and LDLRAD4, which are by far the ‘highest betweenness’ genes (Figure 5B), suggesting a possible role as therapeutic targets.

Among the significantly enriched pathways in the PKH26^+^ state shared by the two lung and colon tissues, we paid particular attention on those related to the process of morphogenesis (Table 6). This focus comes from both the mounting evidence of carcinogenesis as ‘embryo development gone awry’ [37] and the crucial role played by cell morphological changes in cancer and metastatic phenomena [38,39]. In particular, quiescent cancer cells from several tumors have been shown to express increased levels of stemness-associated and embryonic factors [16,22,40,41]. Moreover, quiescent/chemoresistant cells have been shown to adopt a transcriptional program resembling that of embryonic diapause, characterized by c-Myc inactivation and expression of genes typical of the epiblast stage [42,43]. According to these considerations, we focused on the analysis of a morphogenesis module, which was common to QCCs of lung and colorectal cancer.

Going more in detail, for the pathways reported in Table 6, we focused on the genes differentially expressed for quiescent/proliferative states in both lung and colon tissues. For these genes, we built a dataset that reports their expression levels in both lung and colon samples (Appendix A), one that reports the expression levels in both tissues for the PKH26^+^ state only (Appendix A) and one that reports the expression levels in both tissues only for the PKH26^−^ state (Appendix A).

To build the networks, for each dataset and for all pairs of genes in it, the between-genes Pearson correlation coefficient was calculated to define the strength of the gene-gene pairwise connections. We concentrated on gene correlation values, and we adopted the largely accepted r = |0.7| as minimal threshold to consider two genes as correlated, thus establishing an edge between the two genes (nodes) in the resulting interaction network. The characteristics of the resulting networks are summarized in Appendix A. Networks were visualized and analyzed with Cytoscape software (version 3.9.1., Institute for System Biology, Seattle, WA, USA)

Appendix A tells us something interesting: the networks built from the dataset specific for PKH26^+^ (Appendix A) or for PKH26^−^ (Appendix A) have many more edges than the one built on all samples (Appendix A). This suggests that the correlations are not biased by the differential expression between the two PKH26^+^ and PKH26^−^ states (range restriction effect [44]), but refer to constitutive between-genes correlation (intra-class correlations outnumbering entire data set correlations) and thus can be considered as proxies of actual regulatory co-expression circuits.

To get a graphical picture of the networks, we adopted and extremely elevated threshold (r > |0.98|) for two nodes (genes) to be considered as connected. This near to unity threshold indicates the ‘obligated’ relations between gene couples, thus highlighting the ‘hard wired’ modules in morphogenesis. Figure 6B–D reports the obtained graphs. It is immediate to note the presence of a ‘giant connected component’ collecting the majority of obligated contacts in both PKH26^+^ (B) and PKH26^−^ (C), while the all-samples network is much less dense (D). This giant component corresponds to the ‘core’ of the morphogenesis network.

Figure 6A highlights that the high-betweenness genes (i.e., those more involved in long-range regulation and communication between different network domains) are different in the two cases. The fact that different genes play the role of ‘strong’ connectors in the two gene expression network points to a rewiring of the network going from PKH26^+^ and PKH26^−^. This rewiring implies a local ‘role exchange’ of single genes that keeps invariant the average betweenness of the two networks (Appendix A). In this case we considered as connection the usual r > |0.7| threshold.

The Clustering coefficient (Appendix A) has to do with the edge density in the neighborhood of a node, and to be more specific, it corresponds to the proportion of ‘complete triplets’. That is to say the ratio between the number of times in which holds the rule ‘IF an edge exists between node(i) and node(j) AND an edge exists between node(j) and node(k) THEN must exist an edge between node(i) and node(k)’ and the total number of triplets connected by at least two edges. The near to unity mean values of clustering centrality have to do with very high density of edges of the co-expression networks (that in turn derives from the need of a strict coordination of morphogenesis processes), again the higher Clustering coefficient of PKH26^−^ has to do with the higher number of edges. Betweenness analysis (Figure 6A) indicates possible targets specific for quiescent cells (it is worth noting the neat discrimination between elevated centrality nodes relative to PKH26^+^ and PKH26^−^ conditions). As a matter of fact, high centrality corresponds to a major role of the node in signal spreading across the network and thus is a proxy for the character of a gene as a potential target. Passing to the deterministic (r > |0.98|) core of morphogenesis module (Figure 6B–E), Figure 6E suggests that 49 genes of the giant components make up an irreplaceable module of deterministic behavior, while 40 and 61 genes characterize the PKH26^+^ and PKH26^−^ classes, respectively.

Looking at the pathways corresponding to the genes composing the different classes (Table 7), it emerges that the distribution of the pathways seems not to vary across the different classes. This means that the genes across these classes participate with interchangeable roles to the ‘deterministic’ core of the network. This finding is confirmed by the correlation matrix (Table 8) for classes reported in Table 7. Therefore the involved genes have superimposable biological functions and the role played by a gene in one class can be carried out by a different one in the other (Figure 5A, Table 7 and Table 8).

All in all, morphogenesis appears as a highly structured module, constituted by genes strongly interacting among them in a quasi-deterministic way. This is consistent with the presence of strongly interconnected gene regulation networks supporting the activation of atavistic (unicellular-like) development programs in [38,39].

## 3. Discussion

Understanding quiescence-associated transcriptional programs is essential to devise more effective strategies for cancer treatment, as QCCs are implicated in tumor chemoresistance, dissemination and dormancy/reawakening [3]. Previous efforts were made to generate a quiescence-associated signature through the cross-analysis of transcriptomes in normal quiescent stem cells, such as hematopoietic stem cells, muscle stem cells and hair follicle stem cells [45]. By contrast, cancer quiescence is much less understood and the possibility to identify shared quiescence-associated targets among different tumors is largely unexplored. We compared the gene expression profiles of QCCs isolated from NSCLC and CRC tumor xenografts, thus identifying a core quiescence-associated transcriptional program common to both tumors. From a methodological point of view, QCCs isolation performed in this study was characterized by two distinctive features. First, QCCs were isolated ex vivo from tumor xenografts, thus reproducing the features of QCCs present in human tumors with increased faithfulness as compared to in vitro cultured cells. In fact, tumor xenografts were generated by inoculating primary tumor spheroids (which have been reported to be enriched in CSCs and to retain several properties of parental tumors [24,46]) stained with PKH26. Label-retaining/PKH26^+^ cells were then isolated from established xenografts, thus providing the possibility to identify cells that were quiescent/slow cycling from the initial stages of tumor development (“historically quiescent cells”) and not just in a given moment, as occurs instead with the H2B-GFP system or with cell cycle-based reporters. Supporting the soundness of this experimental method, PKH26^+^ QCCs populations were previously isolated from CRC, lung cancer, breast cancer, glioblastoma and osteosarcoma [16,18,20,21,22,29,47]. Importantly, in all the studies cited above, the PKH26^+^ subpopulation was demonstrated to possess stemness traits. According with these findings, a specific search for stemness-related genes in the PKH26^+^ subset revealed an increased expression of the pluripotency factor KLF4 that was common to NSCLC and CRC. Interestingly, besides its known role in pluripotency determination [48,49], KLF4 has been reported to inhibit cancer cell proliferation both in NSCLC and in CRC [50,51,52]. Based on these observations, it may be speculated that KLF4 may play a double role in the maintenance of the QCCs pool by maintaining a stemness state and restraining proliferation at the same time. In addition to KLF4, lung QCCs were characterized by an increased expression of urokinase plasminogen activator surface receptor (uPAR), a component of the plasminogen activation system involved in tumorigenesis, metastasis, EMT and chemoresistance [53], and by the two stemness-associated surface markers CD44 and CD166. Similarly, colorectal QCCs showed an increased expression of factors involved in stemness and self-renewal such as AXIN2, LGR5 and BMI1. Among these, BMI1 was previously identified as the marker for a subpopulation of quiescent intestinal stem cells (ISCs) [54], whereas LGR5 can be found on both proliferating and slow cycling/chemoresistant ISCs [55]. Among the highest-ranking genes expressed in lung QCCs, we also found ZEB2 (Zinc finger E-box-binding homeobox 2), a transcriptional inhibitor involved in EMT, stem cell plasticity and chemoresistance [56,57]. As ZEB2 was previously implicated in quiescence/chemoresistance in CRC [22], its involvement also in NSCLC QCCs suggests a broad role of this factor in regulating cancer quiescence. Altogether, these conditions provided a robust experimental system for the investigation of quiescence-associated features that was validated at several levels during data analysis. In fact, the existence of a PCA component allowing for a perfect partition of the loading component space into PKH26^+^ and PKH26^−^ areas for each tumor represents an intrinsic quality control of samples and suggests the presence of shared gene-regulatory networks supporting a highly organized autonomous dormancy attractor. Furthermore, as discussed in detail below, genes upregulated in NSCLC and CRC QCCs were coherent with a known quiescence-associated phenotype characterized by an elevated expression of pro-EMT and pro-metastatic factors. Specifically, the analysis of genes upregulated in both NSCLC and CRC quiescent cells showed a “core quiescence program” including approximately 11.5% of genes upregulated in lung cancer and 22% of genes upregulated in CRC. Among the 688 genes common to NSCLC and CRC QCCs, we found an enrichment in gene categories involved in angiogenesis, cancer stemness and TGF-β signaling and several factors involved in metastatization (Cathepsin A, ROCK2), hypoxia regulation (EGLN1) and in the ubiquitin proteasome system (Cul3, Cul4). GSEA analysis of gene expression profiles further revealed a common involvement of pathways related to EMT, TGF-β, chemotaxis and cell adhesion in NSCLC and CRC quiescent cells. Factors involved in the negative regulation of cell proliferation were also more represented in QCCs, providing an internal control of population quality. A large body of evidence has previously associated EMT with chemoresistance [58]. Moreover, several studies highlighted the existence of aggressive cancer cells characterized by combined properties of EMT, quiescence and stemness in breast and colorectal tumors [22,59,60,61]. Likewise, TGF-β pathway activation was shown to be implicated in cancer cell quiescence and metastasis [62], in line with its conspicuous presence in the “core quiescence program” shared by NSCLC and CRC. The presence of cell adhesion pathways in the shared QCCs signature is also in line with previous studies showing a central role of adhesion molecules in maintaining cancer dormancy [63] and in protecting dormant cells from chemotherapy through interaction with the perivascular niche [64]. By contrast, the role of chemotaxis in cancer quiescence is less understood. However, deregulated chemokines produced by tumor cells have been shown to drive chemotaxis of immune cells into the tumor and to manipulate them in order to promote cancer cell dissemination [65]. By contrast, rapidly proliferating cells exhibited a higher expression of pathways involved in RNA processing, in line with their increased RNA content and transcriptional activity. Importantly, Myc targets were found to be upregulated in rapidly proliferating cells, in line with Myc activation as a turning point for quiescent and proliferative states in embryos and cancer [42,66]. Proliferating cells also showed a common upregulation of mitosis-associated pathways, in accordance with their replicative phenotype. The dormancy attractor was analyzed in depth in the case of morphogenesis module, confirming the presence of a coherent gene expression interaction pattern. Network analysis of the morphogenesis module shown in Figure 5A highlights different ‘high centrality’ nodes (genes) for the PKH26^+^ and PKH26^−^ states. Such genes are of utmost importance for therapeutic intervention, as they represent crucial nodes for intracellular signaling and provide a distinctive vulnerability profile for quiescent and proliferating cancer cells. Importantly, the key network nodes (high betweenness) for the quiescent subpopulation are represented by Annexin A1, a protein involved in metastasis, suppression of inflammation and of the immune response [67,68,69,70,71], by the antioxidant enzyme catalase, involved in the protection from reactive oxygen species and in chemotherapy resistance [72], and by Methaderin, a protein involved in multiple steps of tumor progression and metastasis [73]. It is worth noting these genes emerged by a pure data-driven strategy on the entire gene expression profiles.

Altogether, our results provide for the first time a core quiescence program shared between lung and colorectal tumors, highlighting the upregulation of pro-metastatic, EMT-like, stemness and chemoresistance pathways in QCCs. Moreover, network analysis of QCCs-associated morphogenesis module revealed new factors that represent crucial nodes in QCCs signaling networks, indicating potential targets for novel QCCs-directed therapeutic approaches.

## 4. Materials and Methods

### 4.1. Primary Non-Small Cell Lung Cancer and Colorectal Cancer Cells

NSCLC and CRC cells were isolated as previously described [15,18,19,20] from surgically resected tumor samples in accordance with the standards of the ethics committee on human experimentation of the Italian National Institute of Health (Istituto Superiore di Sanità, ISS authorization no. CE5ISS 09/282) and subsequently stored at the institutional Cancer Stem Cell Biobank. Tumor cells were cultured as multicellular spheroids in serum-free medium containing epidermal growth factor 20 ng/mL and basic fibroblast growth factor 10 ng/mL (PeproTech, Cranbury, NJ, USA) in nontreated polystyrene flasks (Thermo Fischer Scientific, Waltham, MA, USA). Such culture conditions were used to reduce cell adherence and to support the growth of lung and colorectal cells as stem cell-enriched multicellular spheroids. Regular thawing of early-passage cells was carried out to avoid the accumulation of culture-related changes. Short Tandem Repeats (STR) analysis was performed with the AmpFlSTRIdentifiler Plus Kit (Applied Biosystems, Waltham, MA, USA) and used to generate a unique STR profile for each primary CRC and NSCLC cell line in order to monitor cell purity over time and to confirm matching with the parental tumor. Multicellular spheroid cultures of NSCLC and CRC were routinely tested for their ability to produce carcinomas histologically identical to the parental tumors when injected into NOD.Cg-Prkdcscid Il2rgtm1Wjl/SzJ (NSG) mice (Charles River, Calco, Italy). Cells were routinely tested for mycoplasma contamination with the PCR Mycoplasma Test Kit (PanReac AppliChem GmbH, Darmstadt, Germany).

### 4.2. Xenograft Generation

NSCLC and CRC spheroids were dissociated with TrypLE Express (Thermo Fisher Scientific, Waltham, MA, USA), stained for 2 min at 37 °C with PKH26 (Sigma, St. Louis, MO, USA), then washed extensively with PBS. PKH26 staining was assessed by flow cytometry and cells were used for subsequent experiments only when positivity was ≥98%. NOD.Cg-Prkdcscid Il2rgtm1Wjl/SzJ (NSG) mice (The Jackson Laboratory, Sacramento, CA, USA) were injected subcutaneously with 5 × 10^5^ cells derived from the multicellular spheroid culture of primary tumor specimens (as described above) resuspended in 100 μL 1:1 PBS/Matrigel (BD Biosciences, San Jose, CA, USA). After 3 weeks (for colorectal cancer xenografts) or 4 weeks (for lung cancer xenografts) of tumor growth, animals were euthanized, and tumors were extracted for subsequent sorting of PKH26^+^/PKH26^−^ cells. The time for PKH26^+^ cell isolation was determined through previous experiments and corresponded to a PKH26^+^ positivity of approximately 5%. All animal procedures were performed according to the Italian National Animal Experimentation Guidelines (D.L.116/92) upon approval of the experimental protocol by the Italian Ministry of Health’s Animal Experimentation Committee (DM n. 292/2015 PR 23/4/2015).

### 4.3. Sample Preparation

Xenografts derived from PKH26-stained cells were cut into small pieces and subsequently digested with TrypLE express for 15 min at 37 °C with vigorous pipetting every 5 min. Freshly isolated cells were stained with anti-human EpCAM antibody and 10 μg/mL 7-aminoactinomycin D was used for dead cell exclusion. Cells were sorted with a FACSAria (Becton Dickinson, Franklin Lakes, NJ, USA) to obtain EpCAM^+^/PKH26^+^ and EpCAM^+^/PKH26^−^ populations.

### 4.4. Transcriptional Profiling of NSCLC and CRC Quiescent and Proliferating Cells Derived from Human Xenografts

The transcriptome of PKH26^+^ and PKH26^−^ colorectal cancer cells was obtained as described in [22]. For lung cancer cells, PKH26^+^ and PKH26^−^ cells were isolated from NSCLC xenografts in triplicate samples and subsequently analyzed by RNA-sequencing on the Illumina HiSeq platform as 2 × 150 bp paired-end reads by the company Biodiversa (Rovereto, Italy). RNA libraries were prepared using the SMARTer Stranded Total RNA-Seq Kit v2 (Takara Bio, San Jose, CA, USA) technology that enables removal of ribosomal cDNA following cDNA synthesis (as opposed to direct removal of corresponding rRNA molecules prior to reverse transcription). An average of 98 Mb reads per library was obtained. Trimmomatic [74] was used to remove sequence adapters, discard low-quality reads and trim poor-quality bases. Transcript abundance estimation was performed by means of Salmon 1.3.0 [75], which accurately quantify transcripts without the need of previously aligning the reads, using NCBI hg19 RefSeq reference transcripts. The overall average mapping rate was 14.5%. We performed a series of quality controls in order to understand the reason of such a low mapping rate (we checked for the presence of rRNA, tried both a more aggressive quality trimming of the reads and ran Salmon with different parameters, tried switching reference to GRCh38 and mapping reads to all the genome with STAR aligner and calculated multiple metrics using Picard CollectRnaSeqMetrics and RSeQC), but none of these approaches incremented the mapping rate. Therefore, we infer that the low mapping rate was likely due to the small amount of starting material, as PKH26^+^ cells isolated from tumor xenografts after 3–4 weeks of growth are few. Nonetheless, the percentage of remaining reads were high enough to produce results that were reliable and very consistent among lung and colorectal samples.

### 4.5. Statistical and Bioinformatics Analysis

Gene expression profiles were analyzed by means of Principal Component Analysis so as to select a specific component that neatly discriminates quiescent and proliferating samples in the loading space. PCA was performed in R using the ‘prcomp’ function of the ‘stats’ library. The score distribution of the discriminating component was then used to select the genes most involved in the component (score > |3|). This ‘PCA-derived signature’ was then compared with the signature coming from an independent approach and the signature coming from a standard RNA-seq analysis tool (*limma* trend analysis) obtaining a very strong superposition. The genes coming from *limma* approach were in turn assigned to different putative pathways and functions by GSEA software. The morphogenesis module coming from GSEA was the basis of the complex network approach. Adopting a correlation threshold of Pearson r > |0.7| for the insertion of an edge between two nodes (genes) of the morphogenesis GRN, we generated the networks relative to the quiescent and proliferative states. These networks were analyzed by Cytoscape software in terms of their relative density (frequency of connections) and, on a gene-by-gene basis, for the centrality of different nodes adopting the betweenness centrality index. The identification of the ‘core’ of the GRNs was obtained by the setting of the correlation threshold to r > |0.98|.

## Figures and Tables

**Figure 1 ijms-23-09869-f001:**
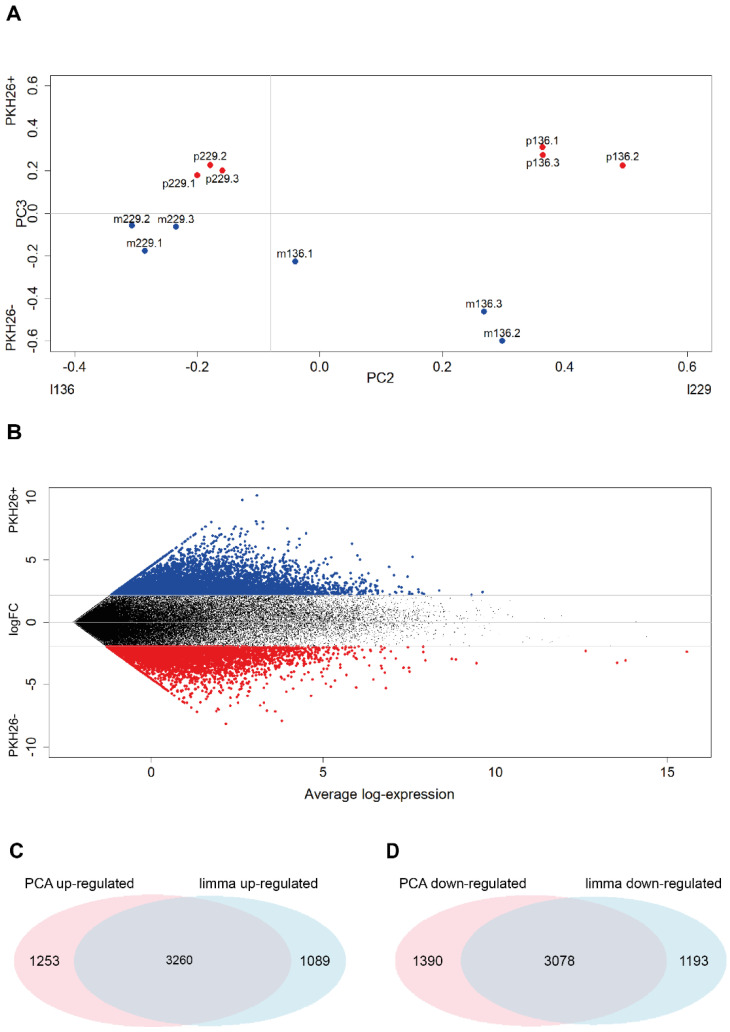
PCA and *limma* results. (**A**) Samples projection on the loading space of the second and third components. Each point corresponds to a sample of the log-transformed original expression data matrix, whose coordinates are given by the PC2 and PC3 loadings of that sample (see Table 1B). It is worth noting the cell line discrimination on PC2 and the quiescent/proliferative separation on PC3. (**B**) Mean difference plot reporting average log expression values versus log2 fold changes (logFC). LogFC are obtained performing differential expression analysis on the log-transformed original expression data matrix. Significantly differentially expressed genes are obtained by sorting in decreasing order the LogFC distribution and keeping the top 6000 values (highlighted in red—PKH26^+^ specific) and the bottom 6000 (highlighted in blue—PKH26^−^ specific), which resulted to be associated to |LogFC| > 1.95. (**C**,**D**) Venn diagrams reporting the number of significantly up-regulated genes in the PKH26^+^ state identified by the *limma* and PCA approaches highlighting the relevant superposition between the two methods. Duplicated genes are removed. Statistical significance of the overlap: (**C**) *p*-value < 1 × 10^−4^; RR = 24.0; (**D**) *p*-value < 1 × 10^−4^; RR = 23.4.

**Figure 2 ijms-23-09869-f002:**
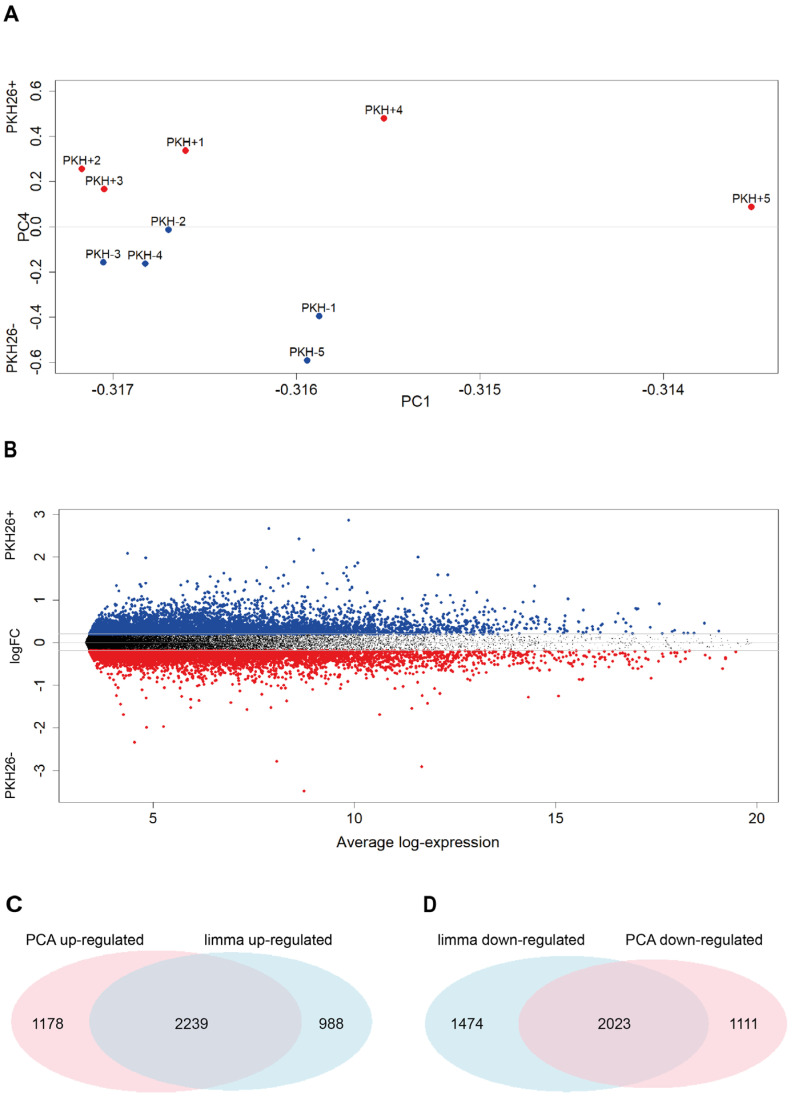
Differential expression analysis performed with Principal Component Analysis is consistent with that performed with the standard *limma*. (**A**) Samples projection on the loading space of the first and fourth components. Each point corresponds to a sample of the log-transformed original expression data matrix, whose coordinates correspond to PC2 and PC3 loadings of that sample (see Table 2B). (**B**) Mean Difference plot reporting average log expression values versus log2 fold changes (logFC). LogFC are obtained performing differential expression analysis on the log-transformed original expression data matrix. Significantly differentially expressed genes are achieved sorting in decreasing order the LogFC distribution and keeping the top 6000 values (highlighted in red—PKH26^+^ specific) and the bottom 6000 (highlighted in blue—PKH26^−^ specific), which resulted to be associated to |LogFC| > 0.2. (**C**,**D**) Venn diagrams reporting the number of significantly up-regulated genes in the PKH26^+^ state identified by the *limma* and PCA approaches highlighting the relevant superposition between the two methods. Duplicated genes were removed. Statistical significance of the overlap: (**C**) *p*-value < 1 × 10^−4^; RR = 12.9; (**D**) *p*-value < 1 × 10^−4^; RR = 11.8.

**Figure 3 ijms-23-09869-f003:**
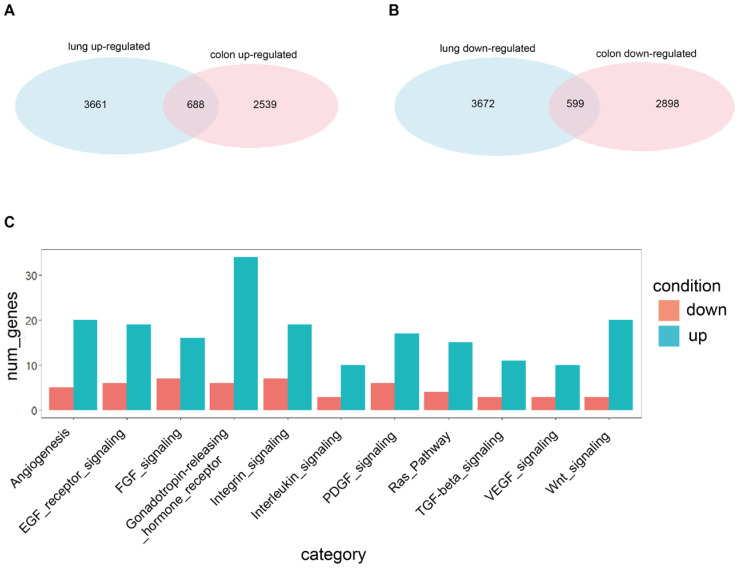
Shared genes resulting from differential expression analysis performed with the standard *limma*. (**A**) Venn diagram reporting the number of significantly up-regulated genes in the PKH26^+^ state identified by the *limma* approach in the lung (3661 genes) and colon (2539 genes) systems, and in both systems (688 genes). Statistical significance of the overlap: *p*-value < 1 × 10^−4^; RR = 7.2. Genes are considered significant if they appear in the first 6000 of the ranking built on the logFC, with a nominal *p*-value < 0.05. Duplicated genes are removed. (**B**) Venn diagram reporting the number of significantly down-regulated genes in the PKH26^+^ state identified by the *limma* approach in the lung (3672 genes) and colon (2898 genes) systems and in both systems (599 genes). Statistical significance of the overlap: *p*-value < 1 × 10^−4^; RR = 5.9. Genes are considered significant if they appear in the last 6000 of the ranking built on the logFC, with a nominal *p*-value < 0.05. Duplicated genes are removed. (**C**) Barplot reporting the number of significantly up- and down-regulated genes in the PKH26^+^ state, identified by the *limma* approach and shared by the lung and colon systems, belonging to some interesting KEGG categories (reported on the *x*-axis) found using Panther. Abbreviations: num_genes = number of genes; down = down-regulated; up = up-regulated.

**Figure 4 ijms-23-09869-f004:**
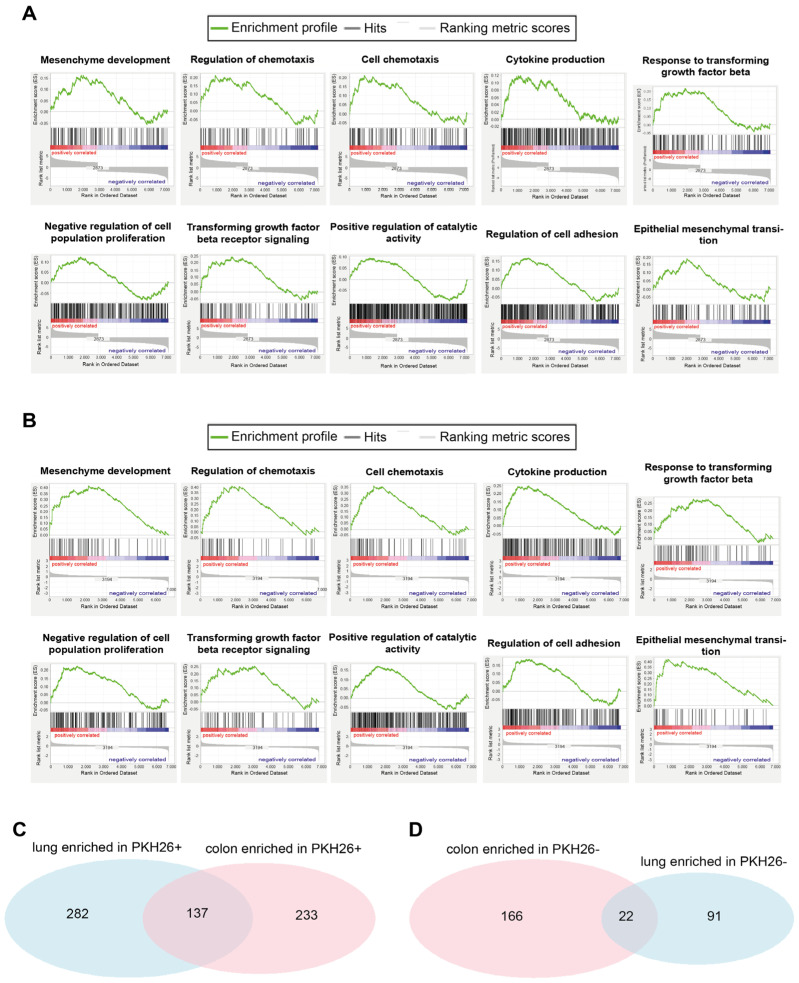
Lung QCCs and colorectal QCCs share some interesting significantly enriched pathways. (**A**) RNA sequencing-generated gene set enrichment analysis for Mesenchyme Development (nominal *p*-value = 0.032, FDR *q*-value = 0.212), regulation of chemotaxis (nominal *p*-value = 0.012, FDR *q*-value = 0.127), cell chemotaxis (nominal *p*-value = 0.005, FDR *q*-value = 0.105), cytokine production (nominal *p*-value = 0.050, FDR *q*-value = 0.271), response to transforming growth factor beta (nominal *p*-value < 1 × 10^−10^, FDR *q*-value = 0.058), negative regulation of cell population proliferation (nominal *p*-value = 0.018, FDR *q*-value = 0.248), transforming growth factor beta receptor signaling (nominal *p*-value < 1 × 10^−10^, FDR *q*-value = 0.057), positive regulation of catalytic activity (nominal *p*-value = 0.019, FDR *q*-value = 0.256), regulation of cell adhesion (nominal *p*-value < 1 × 10^−10^, FDR *q*-value = 0.064) and epithelial mesenchymal transition (nominal *p*-value = 0.005, FDR *q*-value = 0.051) in PKH26^+^ lung cancer cells (compared with PKH26^−^ cells). (**B**) RNA sequencing-generated gene set enrichment analysis for Mesenchyme Development (nominal *p*-value < 1 × 10^−10^, FDR *q*-value = 0.015), regulation of chemotaxis (nominal *p*-value < 1 × 10^−10^, FDR *q*-value = 0.026), cell chemotaxis (nominal *p*-value = 0.002, FDR *q*-value = 0.038), cytokine production (nominal pvalue = 0.002, FDR *q*-value = 0.069), response to transforming growth factor beta (nominal *p*-value = 0.011, FDR *q*-value = 0.108), negative regulation of cell population proliferation (nominal *p*-value = 0.002, FDR *q*-value = 0.105), transforming growth factor beta receptor signaling (nominal *p*-value = 0.046, FDR *q*-value = 0.178), positive regulation of catalytic activity (nominal *p*-value = 0.030, FDR *q*-value = 0.230), regulation of cell adhesion (nominal *p*-value = 0.047, FDR *q*-value = 0.250) and epithelial mesenchymal transition (nominal *p*-value < 1 × 10^−10^, FDR *q*-value = 0.002) in PKH26^+^ colon cancer cells (compared with PKH26^−^ cells). (**C**) Venn diagram reporting the number of significantly enriched gene sets identified in lung PKH26^+^ cells (282 enriched pathways, *p*-value < 0.05), colon PKH26^+^ cells (233 enriched pathways, *p*-value < 0.05) and in both lung and colon PKH26^+^ cells (137 enriched pathways, *p*-value < 0.05). Statistical significance of the overlap: *p*-value < 1 × 10^−4^; RR = 3.4. (**D**) Venn diagram reporting the number of significantly enriched gene sets identified in lung PKH26^−^ cells (91 enriched pathways, *p*-value < 0.05), colon PKH26^−^ cells (166 enriched pathways, *p*-value < 0.05) and in both lung and colon PKH26^−^ cells (22 enriched pathways, *p*-value < 0.05). Statistical significance of the overlap: *p*-value = 0.003; RR = 4.4.

**Figure 5 ijms-23-09869-f005:**
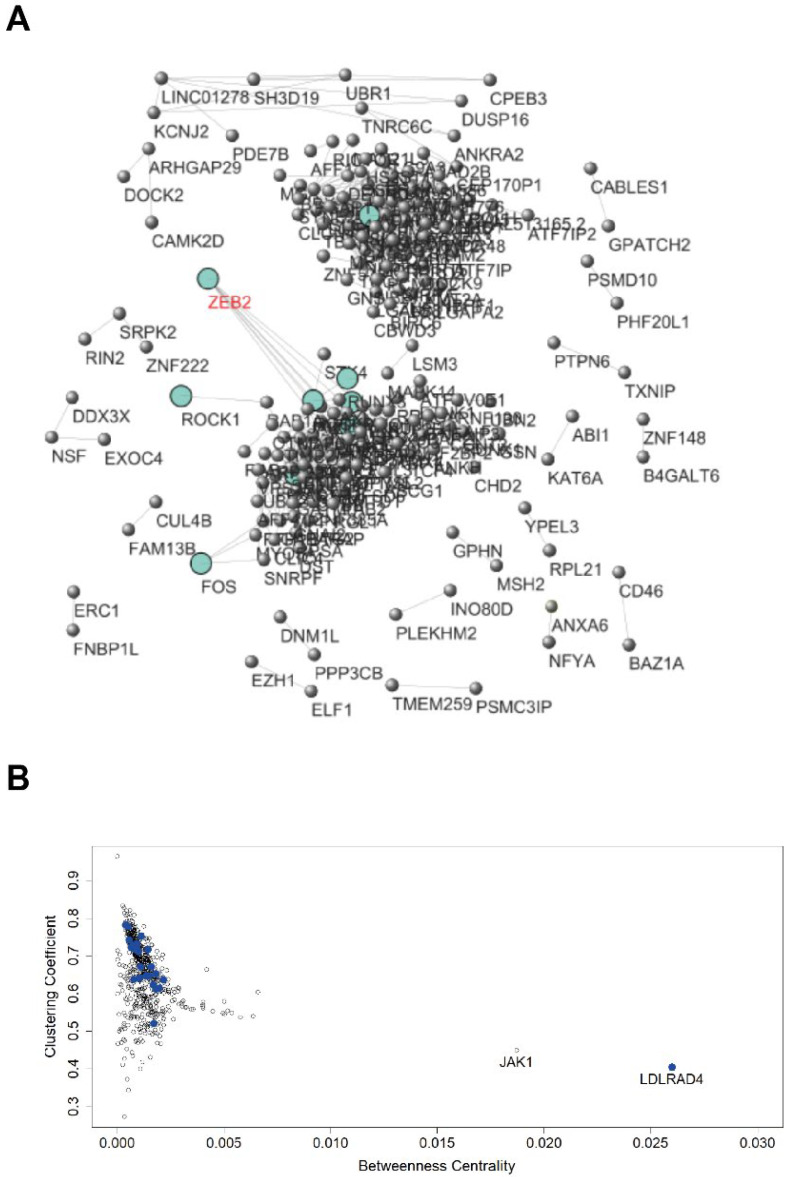
Network built on the genes up-regulated in PKH26^+^ shared by the lung and colon systems. (**A**) Network visualization. It was built considering as connected only pairs of nodes with extremely elevated correlations (r > |0.98|). (**B**) Genes’ projection in the Clustering coefficient/betweenness space. It refers to the network built considering the usual threshold of Pearson r = |0.7|. Genes linked to the TGF-β pathway are highlighted in blue.

**Figure 6 ijms-23-09869-f006:**
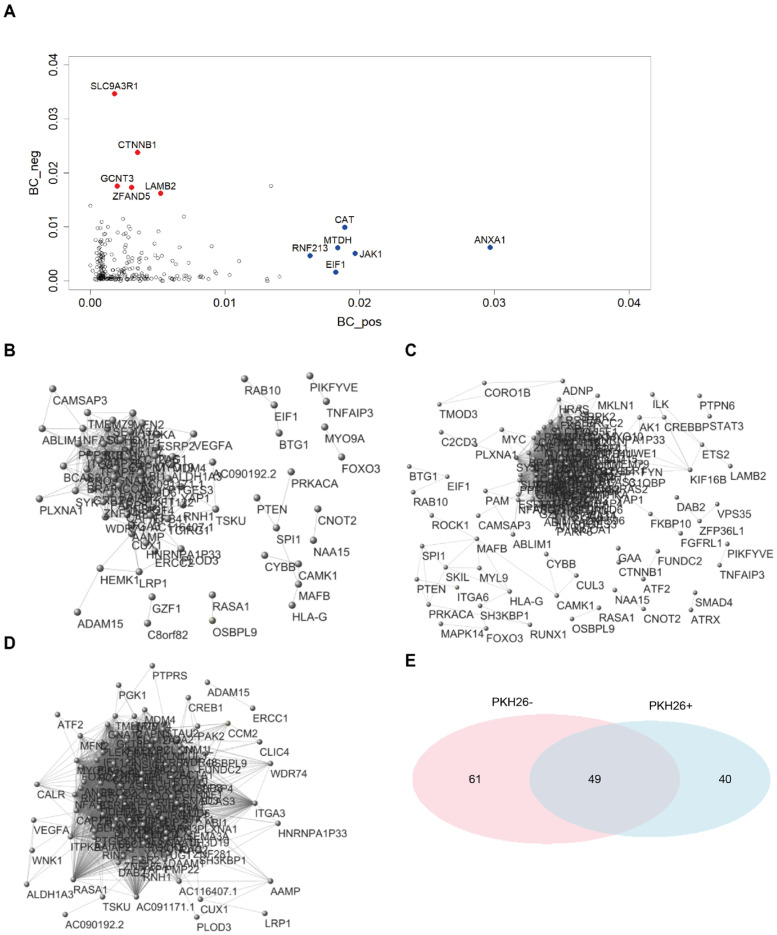
Network built on the PKH26^+^, PKH26^−^ and all-samples datasets. (**A**) Betweenness centrality scores in both PKH26^+^ (*x*-axis) and PKH26^−^ (*y*-axis) networks. “High traffic” genes behaving differently between PKH26^+^ and PKH26^−^ are highlighted in red. (**B**) Network visualization from the PKH26^+^ dataset. Network was built considering as connected only pairs of nodes with extremely elevated correlations (r > |0.98|). See Appendix A for more details. (**C**) Network visualization from the PKH26^−^ dataset. Network was built considering as connected only pairs of nodes with extremely elevated correlations (r > |0.98|). See Appendix A for more details. (**D**) Network visualization from the all-samples dataset. Network was built considering as connected only pairs of nodes with extremely elevated correlations (r > |0.98|). See Appendix A for more details. (**E**) Venn diagram reporting the number of genes identified in the PKH26^+^ (40 genes) and in the PKH26^−^ network (61 genes), and in both PKH26^+^ and PKH26^−^ networks (49 genes). Networks were built considering as connected only pairs of nodes with extremely elevated correlations (r > |0.98|). Statistical significance of the overlap: *p*-value < 1 × 10^−4^; RR = 3.6.

**Table 1 ijms-23-09869-t001:** PCA results. (**A**) Decomposition of the correlation matrix expressed as eigenvalues, difference between adjacent eigenvalues, proportion of variance explained by each component and cumulative variance. (**B**) Component pattern (loadings). Each loading corresponds to the correlation coefficient between original variables (samples) and corresponding component.

**(A)**				
**Component**	**Eigenvalue**	**Difference**	**Proportion**	**Cumulative**
1	5.970	4.816	0.498	0.498
2	1.154	0.248	0.096	0.594
3	0.906	0.307	0.076	0.670
4	0.599	0.049	0.050	0.720
**(B)**				
**Sample**	**PC1**	**PC2**	**PC3**	**PC4**
p136.1	0.273	0.364	0.300	0.053
p136.2	0.230	0.495	0.224	−0.037
p136.3	0.275	0.364	0.275	0.068
m136.1	0.305	−0.040	−0.225	0.358
m136.2	0.213	0.298	−0.600	−0.689
m136.3	0.246	0.268	−0.462	0.531
p229.1	0.320	−0.200	0.178	−0.160
p229.2	0.318	−0.178	0.226	−0.166
p229.3	0.316	−0.159	0.201	−0.140
m229.1	0.318	−0.285	−0.176	0.133
m229.2	0.325	−0.307	−0.052	0.068
m229.3	0.296	−0.235	−0.064	−0.113

**PC1** = ‘tissue attractor’ [30]. This component is by far the most relevant in terms of percent of variance explained (around 50%) and has all positive loadings; thus, it can be considered as a ‘size’ component [31] stemming from the largely invariant tissue-specific average genome expression profile. **PC2** = ‘cell line effect’ this component accounts for approximately 10% of total variance; it is a ‘shape’ component [31] having both positive and negative loading correspondent to a directionality of effect having as poles (opposite loadings, see Table 1B) the lsc136 and lsc229 cell lines. This implies the presence of a neat ‘effect line’ (no exception to this loading sign rule, Table 1B) pointing to the presence of genes with a correlated expression with opposite up (down) expression in the two lines independently of the PKH26^+^/PKH26− phenotype. **PC3** = ‘quiescent/replicative effect’ this component accounts for 7.5% of total variance and discriminates PKH26^+^ and PKH26^−^ (p vs. m) phenotypes in terms of loading sign. Again, this is a shape component pointing to the presence of a clear opposite pattern of gene activation/repression discriminating the two conditions. It is worth noting that principal components are each other orthogonal by construction, thus implying the above effects (tissue attractor, line effect, phenotype effect) do not superimpose. This means that the quiescent/dormant discrimination happens through the same underlying mechanism in both lines.

**Table 2 ijms-23-09869-t002:** Typical ‘stemness’ genes up-regulated in the lung cancer PKH26^+^ state being part of the top 6000 genes selected by *limma* analysis. Differential expression evaluated through *limma* as previously described.

Gene Name	logFC	*p*-Value
KLF4	2.1154	0.0770
PLAUR/CD87	2.3351	0.0837
CD44	3.7835	0.0186
ALCAM/CD166	6.4852	0.0000

**Table 3 ijms-23-09869-t003:** PCA results. (**A**) Decomposition of the correlation matrix expressed as eigenvalues, difference between adjacent eigenvalues, proportion of variance explained by each component and cumulative variance. (**B**) Component pattern (loadings). Each loading corresponds to the correlation coefficient between original variables (samples) and corresponding component.

**(A)**				
**Component**	**Eigenvalue**	**Difference**	**Proportion**	**Cumulative**
1	9.874	9.832	0.987	0.987
2	0.042	0.017	0.004	0.991
3	0.025	0.010	0.003	0.994
4	0.015	0.003	0.002	0.996
**(B)**				
**Sample**	**PC1**	**PC2**	**PC3**	**PC4**
PKH26^+^1	−0.317	0.156	−0.272	0.336
PKH26^+^2	−0.317	0.065	−0.151	0.255
PKH26^+^3	−0.317	0.100	−0.221	0.166
PKH26^+^4	−0.316	−0.038	0.694	0.478
PKH26^+^5	−0.314	−0.799	−0.040	0.087
PKH26^−^1	−0.316	0.347	0.353	−0.396
PKH26^−^2	−0.317	0.143	−0.363	−0.013
PKH26^−^3	−0.317	0.121	−0.262	−0.157
PKH26^−^4	−0.317	0.233	0.208	−0.163
PKH26^−^5	−0.316	−0.336	0.057	−0.591

**Table 4 ijms-23-09869-t004:** Typical ‘stemness’ genes up-regulated in the colorectal cancer PKH26^+^ state. Differential expression evaluated through the *limma* approach.

Gene Name	logFC	*p*-Value
KLF4	0.3838	0.0078
AXIN2	0.4281	0.0481
LGR5	0.8000	0.0095
BMI1	0.3926	0.0015

**Table 5 ijms-23-09869-t005:** Relevant pathways shared by the two lung and colon systems. In yellow are highlighted the pathways enriched in PKH26^−^; in red are highlighted those enriched in PKH26^+^.

Database	Pathway
Hallmark	MYC_TARGETS_V1
Hallmark	MYC_TARGETS_V2
Hallmark	EPITHELIAL_MESENCHYMAL_TRANSITION
Gene Ontology	NCRNA_METABOLIC_PROCESS
Gene Ontology	TRNA_METABOLIC_PROCESS
Gene Ontology	RRNA_METABOLIC_PROCESS
Gene Ontology	MITOTIC_CELL_CYCLE
Gene Ontology	MITOTIC_CELL_CYCLE_PROCESS
Gene Ontology	MRNA_PROCESSING
Gene Ontology	RNA_PROCESSING
Gene Ontology	MESENCHIME_DEVELOPMENT
Gene Ontology	REGULATION_OF_CHEMOTAXIS
Gene Ontology	CELL_CHEMOTAXIS
Gene Ontology	CYTOKINE_PRODUCTION
Gene Ontology	RESPONSE_TO_TRANSFORMING_GROWTH_FACTOR_BETA
Gene Ontology	NEGATIVE_REGULATION_OF_CELL_POPULATION_PROLIFERATION
Gene Ontology	TRANSFORMING_GROWTH_FACTOR_BETA_RECEPTOR_SIGNALING_PATHWAY
Gene Ontology	POSITIVE_REGULATION_OF_CATALYTIC_ACTIVITY
Gene Ontology	REGULATION_OF_CELL_ADHESION

**Table 6 ijms-23-09869-t006:** Morphogenesis pathways, significantly enriched in the PKH26^+^ state, shared by the two lung and colon systems.

Database	Pathway
Gene Ontology	ANATOMICAL_STRUCTURE_FORMATION_INVOLVED_IN_MORPHOGENESIS
Gene Ontology	ANIMAL_ORGAN_MORPHOGENESIS
Gene Ontology	BLOOD_VESSEL_MORPHOGENESIS
Gene Ontology	EMBRYONIC_MORPHOGENESIS
Gene Ontology	HEART_MORPHOGENESIS
Gene Ontology	MORPHOGENESIS_OF_AN_EPITHELIUM
Gene Ontology	REGULATION_OF_ANATOMICAL_STRUCTURE_MORPHOGENESIS
Gene Ontology	TISSUE_MORPHOGENESIS
Gene Ontology	TUBE_DEVELOPMENT
Gene Ontology	TUBE_MORPHOGENESIS

**Table 7 ijms-23-09869-t007:** Pathway distribution across the classes. PKH26^+^, PKH26^−^ and all-samples datasets refer to the genes composing the giant components reported in Figure 6B–E.

Pathway	PKH26^+^	PKH26^−^	All_Samples
ANATOMICAL_STRUCTURE_FORMATION_INVOLVED_IN_MORPHOGENESIS	16	28	18
ANIMAL_ORGAN_MORPHOGENESIS	17	23	15
BLOOD_VESSEL_MORPHOGENESIS	7	17	7
EMBRYONIC_MORPHOGENESIS	7	12	8
HEART_MORPHOGENESIS	7	4	1
MORPHOGENESIS_OF_AN_EPITHELIUM	9	12	6
REGULATION_OF_ANATOMICAL_STRUCTURE_MORPHOGENESIS	12	24	14
TISSUE_MORPHOGENESIS	12	15	8
TUBE_DEVELOPMENT	14	23	11
TUBE_MORPHOGENESIS	11	22	9

**Table 8 ijms-23-09869-t008:** Correlation matrix for Table 5.

	PKH26^+^	PKH26^−^	All_Samples
**PKH26^+^**	1.00	0.85	0.93
**PKH26^−^**	0.85	1.00	0.80
**all_samples**	0.93	0.80	1.00

## Data Availability

The original contributions presented in the study are included in the article/Appendix A. Further inquiries can be directed to the corresponding authors.

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
