# Peer review of "Analysis of Dormancy-Associated Transcriptional Networks Reveals a Shared Quiescence Signature in Lung and Colorectal Cancer"

_ijms, 2022, doi:10.3390/ijms23179869_

Round 1
Reviewer 1 Report
Likewise, many tumors contain subpopulations of quiescent cells with features of stem cells involved in clinically relevant steps of tumor development such as chemoresistance, metastatic dissemination, dormancy, and relapse. The authors should try to distinguish between cancer stem cells and persister cells since these have different definitions and not all functions mentioned are linked to cancer stem cells. Also how these two populations compare to dormant/ quiescent cells has not been explained.
The precise definition of quiescent or slow-cycling tumour cells in these two cancers is essential and the authors could clearly specify their markers and isolation criteria in lines 62 or so.
From a more theoretical perspective, we analysed the morphogenesis module in quiescent and proliferating cells of both tumors, pointing to relevant differences in gene regulatory network structure. Please specify and explain what gene network you refer to.
A wiring architecture of gene expression correlation for quiescent cells states more fit to a metastable state like dormancy with respect to actively proliferating cells. It would be interesting to know what you mean by a metastable state.
Methods: Please provide the method used for dissecting and isolating these cells, albeit briefly (line 464). For example, what markers were used to distinguish between quiescent tumour cells and cancer stem cells and if in the opinion of the authors these are the same cells or different. Its key that the authors can show their isolation method is accurate (I am aware some previous methods have been cited) but please reexplain your methods precisely since the definition and characterisation of the intended population can affirm or disprove the results obtained. From what the authors have written the population seems to have CSC characteristics such as spheroid formation yet the characteristics of dormant cells in the literature outline their low ki-67 profile, p38MAPK/ ERK ratios, and the retention of PKH26 dye due to slow cycles, so it is interesting to know how the authors isolate them and how much this population overlaps with CSCs (with respect to surface markers and general characteristics), so far this manuscript does not give this reassurance.
Again, the ability to produce new carcinomas in line 475, is the characteristic of cancer-initiating cells not CSCs nor quiescent cells, quiescence can be a characteristic of a CSC but whether they are one and the same is not that clear. In actual fact, the literature mixes and matches these cells and many define these cells and their characteristics differently but the authors don’t take a strong stance on defining these cells and overlapping cell populations. Dormant cells when xenografted to mouse models, should produce indolent lesions and not readily disseminate as would high proliferate tumour cells, hence forming tumour proper in a xenograft model makes me think that cancer-initiating cells were also included in the PKH26+ cell population. What other markers and criteria were used to purify quiescent cells? Is this marker (PKH26+) sufficient to isolate quiescent cells from all tissues and could some low-cycling non-tumour cells be selected as well? In what disease stage were the patients when biopsies were taken, was this post-treatment? Why was quiescence induced in these cells, were the cells under selective pressure?
In method 4.3. Again, I wouldn’t expect quiescent cells to produce a palpable tumour in vivo, so what type of lesions were formed by PKH26+ cells, and how these were detected in vivo and isolated. Also, upon histological analysis of this xenograft what was observed? For example, heavy collagen and fibronectin presence, rich CD31+ vasculature, low ki-67 dye, etc? Until these questions are answered, it will be difficult to confidently verify the rest of the results.
In figures 1-2, please provide a supplementary sheet containing all the genes featured in this figure. Also please explain what these findings signify in greater depth.
In figure 3, also provide these genes and explains the difference between the gene lists derived from these two methods and what these differences signify.
In figure 3 what would happen if the authors compared the PKH26+/- cell populations, how would the GO terms differ? In table 3 this view is provided but what is so discriminating between these two profiles? It is strange that PKH26+ populations should be linked to EMT since they can only form indolent lesions in vivo...
In figure 4, how was CSCs isolated? What does this data signify about the similarities and differences between CSC and quiescent cells?
I am confused about the findings in table 4 and 7, how do these justify the function of dormant cells?
The content of figure 5 is not legible, please review and amend it.
Reviewer 2 Report
This study focuses on identifying commonalities between two different quiescent cancer cell types—a “quiescence signature”. This is a great premise that constitute important work to understand the biology of quiescence; while the importance of quiescent cells in cancer biology and tumor dormancy is getting recognized over the past decade, there are still few studies that go beyond the characterization of the transcriptome of one type of quiescent cell (or in a tissue) and aim to identify shared features, rather than specific differences. This paper should therefore be of strong interest to the field for researchers working not only on cancer, but also on the fundamental biology of cellular quiescence.
Overall, the paper reads well, although it focuses a bit more on the methods rather than the discussion of the genes comprising the quiescence signature identified by the authors. A nice result is that one of the positive hits was ZEB2, previously identified from CRC by the authors (Francescangeli et al, 2020: ref 15). It would be great if the authors could further describe/explain the biology of their result (cf. remark below), which would considerably strengthen the paper.
Some of the figures should be clarified, especially given the current image resolution:
Fig 1 would be made clearer if points in panel A were also labeled by quiescent/non-quiescent (for example, circle groups of points). The figure legend referring to panel B should also specify if the logFC is log2- or log10-based and what the threshold used was (visually: 2?). The same comments apply to Fig 2, where maybe the resolution of the figure seems to make “-“ symbols invisible (f.e. “PKH+1” ok but “PKH 1” instead of “PKH-1”). A more consistent labeling between Fig 1 and Fig 2 would help. In Fig 2B it looks like the logFC threshold used is quite different, visually ~0.2 or so. Why is that the case? Is this still log2-fold?
Section 2.4 analyzing the common differentially-expressed (DE) genes in lung and colon QCCs should also give an idea of the expected vs observed ratios. The determined gene set was 688 shared genes; what are the odds of this intersection? From Fig 1CD, 3260+3078 are DE in lung QCCs, from 2CD, 2239+2023 are DE. Given a total of x genes, what is the E-value (or equivalent, or ideally a FDR estimate) of finding an overlap of 688+599 or more when intersecting a set of 6338 and a set of 4262 randomly-selected genes? This is a very important measure to estimate the robustness of this approach and the confidence in this group of genes, which will contain false-positives and false-negatives.
line 237: is this GSEA p-value the nominal p-value, or the FWER p-value (i.e. adjusted for multiple comparisons)? (cf. GSEA user guide) – The current low resolution of Figure 4 makes it impossible to read in panels AB. Figure legend 4 indicates that nominal p-values are given; can the authors also indicate FWER-corrected p-values (either here or in an additional table, supplementary or not). Some p-values are given as “<” and would be better given as their actual value: lines 269-264.
For Figure 5, the panels (especially BCD) are un-readable. I would suggest greatly expanding the size of the figure (and resolution!), maybe concentrating on the quiescent network (PKH26+), and/or attaching a high-resolution, high-size image for the others as Supplementary Material. Gene networks are often only informative when gene names can be clearly read on every node (beyond rough topology of the graph).
While Table 5 and Table 6 show interesting network information, they contain data that is usually put in Supplementary Material. I think a more informative use of this space would be to discuss several genes found in the ‘quiescence signature’, such as ZEB2, and/or individual genes found in the interesting networks such as morphogenesis—how many of genes were found to be involved in cellular quiescence before? Is ZEB2 well-connected in the graph? etc. In the current version of the manuscript, this discussion is quite limited (lines 446-451), and the discussed genes are not highlighted in Fig 5, whereas it would be quite interesting and informative to do so (example: use a specific color for TGF-beta genes, another for ZEB2, etc.)
An important point to address also is that the current section 4.3 of the Methods does not include details for the RNA-seq procedure; at the very least, the kit used should be mentioned, whether the RNA was mRNA-enriched or rRNA-depleted. Even if the information is present in ref. 15, this minimal set of information should be provided. Furthermore, the tools used for the bioinformatics analysis should be mentioned and cited (adaptor-trimming? mapping? version of the genome or of the transcriptome?). Furthermore, basic information about the sequencing run should be provided (at the very least, number of reads per library, % mapping), and an accession number (GEO or SRA) should be provided to arrange to put the raw data (fastq files) to a data repository, as per common practice (and often, a required standard for publication in many journals). For section 4.4, it should be mentioned how was the PCA conducted. If, for example, it was done in R, this should be mentioned + which packages were used.
At the first mention of PKH26 (line 84), a brief description of this dye should be provided, at least to explain in the text as well that PKH26 is retained in slow-cycling cells and is therefore used as a marker of quiescence; in the Methods, a brief mention of its mode of action should also be included (lipid membrane marker that gets diluted with cell division)
Minor points/typos:
line 78: “obliged” -> “necessary” or “essential” may be better adapted.
line 84: explain PKH26+/PKH26- ; in line 111 it is PKH26+/-
line 86: “perfect discriminating” needs to be explained
line 94: specify if log2 or log10
line 96: “Rna-seq” is most often “RNA-seq”
“in vitro” and “in vivo” throughout the text would be better if italicized
line 299: maybe an “and” is missing? “We concentrated on gene correlation values and we adopted the (...)”
“between genes correlation” would be clearer as “between-genes correlation”
line 323: here “obliged” may be better as “obligated”
line 513: “betweenness”
Reviewer 3 Report
Title: "Analysis of dormancy-associated transcriptional networks reveals ashared quiescence signature in lung and colorectal cancer”
Alessandro Giuliani, Ann Zeuner
Comments:
The authors of this work have shown that there are common features that characterize QCCs of lung and colorectal cancers and offer new insights into the mechanisms responsible for the regulation of quiescence in solid tumors.
Several Points:
1: Page 1 line 36: "QCC" must also be defined in the introduction before the abbreviation is used
2: Page 1 line 40: here QCC is written out, although the abbreviation is already introduced
3: Page 2 lines 61-73: this is really just describing your methods, results, and what the results show (conclusion); for me, this doesn't belong in the introduction, but in the results, discussion, and conclusion. I also miss a clearly stated overall goal of the study and why it is important in the topic area described.
4: Page 5 Figure 5A: lacks proper labeling of the x and y axes, i.e., what it is about in the first place
5: the same applies to page 7 Figure 2
6: with the individual results, I often miss a sentence what the found implies and what meaning it has. e.g., page 8, line 206/207: what do these results say?
7: Page 9 Figure 3C: the legend "condition" should be written out, probably the "up" and "down" should stand for "up-regulated" and "down-regulated"; also the y-axis with "num_genes" should be fully described as "number of genes" (or similar); also it would be useful to write out the abbreviations/names used for the x-axis in the legend of the figure for better comprehensibility.
8: Page 11 Figure 4A/B: too much information at once, which is also not readable; here a separate overview diagram of the results would be nicer and more descriptive than the representation coming from the machine
9: Page 11 line 290: the heading of the table is in the middle of the text, I also miss a descriptive legend; in the other tables relevant info was at least highlighted in color
10: Page 13 line 318: the word "betwenness" strikes me as very odd (it is used in the rest of the text as well), unless you can use it that way in this "matter"....
11: Page 14 Figure 5: the quality is not sufficient, you can't see anything, especially in Fig. 5B, C, D; also, the descriptive axis labels are missing here too.
12: Table 6, 7, 8: Again, important information is not labeled or it is not highlighted which information is crucial (also not in the corresponding legend).
13: The discussion, like the methods, is of limited evaluability to me because it is extraneous, but I miss a meaningful conclusion at the end of the discussion that lists both strengths and weaknesses/limits and points of contention.
14: the English language/grammar should be improved again in some places.
Round 2
Reviewer 1 Report
Now that the authors have clarified the target population and some other aspects, I think the manuscript is publishable.
Author Response
Thank you.
Reviewer 2 Report
In the revised version of the manuscript, the authors have considerably improved the text and the quality of the figures, addressing most of the comments. The expanded discussion and new network graphs are a clear improvement and easier to read/parse.
Some of these replies should be added to the paper, such as:
- For point (2), the selection of the top/bottom 6000 genes should be explained either in the corresponding panels (Fig 1B, Fig 2B) or at least in the corresponding legends.
- For point (3), estimating the statistical significance of an overlap, should be used to label Venn diagrams in figures (either on the figure themselves below the intersection, or at least in the figure legend), specifically Fig 1CD, Fig 2CD, Fig 3AB and Fig 4CD. (As a note, panel 4C shows 137 genes in overlap, but the text says 136).
Two major issues that have arisen and should be addressed:
- For point (7), the additional experimental details are welcome. An important point remains; the mapping rate of 14.5% is quite low for RNA-seq. What is the reason for this? Can the authors check whether this is due to one of the following possible reasons such as: (i) remaining high amounts of rRNA, (ii) repetitive loci that were not present in the given reference transcriptome set, (iii) the filtering-out of multi-mappers, (iv) parameters used for Salmon?
As the conclusions of the authors will clearly depend on the accuracy of their gene expression data, it would be essential to address this issue (both as a response to this review but then also as a detailed explanation in the manuscript itself).
- It is essential that the accession numbers are included in the manuscript. Many journals require the accession numbers to be working to make a final decision on a manuscript (and oftentimes, to even start the reviewing process).
Minor typos:
line 71: “crossed analysis” -> “cross-analysis”
line 112: “we step” -> “we stepped”
line 395: “betwenness” -> “betweenness”
Reviewer 3 Report
The authors have satisfactorily addressed the concerns raised in the original version. The revised version is significantly improved. No further concerns.
Author Response
Thank you.
Round 3
Reviewer 2 Report
The authors have addressed the previous raised comments in this new revised version of the manuscript, significantly improving it. The additional details in the experimental protocol and discussion are especially welcome, and present in a clear manner the significant result shown in the manuscript: the identification of transcripts commonly expressed in distinct quiescent tumor cells. This will be of strong interest in the field. - For that reason, it is important to insist that it would be a good decision to have accession numbers provided within the manuscript, which had become standard in many fields and is required for most journals. While I understand that some authors may be reluctant to do this, it ensures the open-ness and reproducibility of science and will be beneficial for the paper in the long-run, so I would encourage the authors to consider this.
For point (7) (RNA-seq), the additional comments provide some context to understand the low mapping rate. Given the experimental protocol, a possible explanation is potential background co-isolation of mouse cells despite the EpCAM FACS strategy. An easy way to check for this would be to select all unmapped reads (to hg19), and measure their mapping rate to the Mus musculus genome. I would encourage the authors to check this. On the other hand, the current text provides sufficient explanation for readers.
For the previously raised point (3) (significance of intersections), the authors have added p-values to figure legends. In cases where the p-value is small, an exponential notation would be better as it would allow to give an exact value rather than rounding to zero (for example, p<1×10-20 (?) rather than p=0.000 for legends of Fig 1, 2, 3, 6).
For Fig 5B, along with LDLRAD4, there is another gene that appears to have a high ‘betweenness centrality’ score; it would be worth including its name next to its dot, as for LDLRAD4.
